# U-NET FOR INDOOR PATHLOSS PREDICTION FROM SPARSE MEASUREMENTS WITH PHYSICS-BASED FEATURES

*Khoren Petrosyan* [*][†]    *Rafayel Mkrtchyan* [*][†]
*Hrant Khachatrian* [*][†]    *Theofanis P. Raptis* [‡]

[*]Yerevan State University, Yerevan, Armenia
Email: {khorenpetrosyan, rafayel.mkrtchyan, hrant.khachatrian}@ysu.am
[†]YerevaNN, Yerevan, Armenia
[‡]Institute of Informatics and Telematics, National Research Council, Pisa, Italy
Email: theofanis.raptis@iit.cnr.it

## ABSTRACT

This work was conducted in the context of the MLSP 2025 Sampling-Assisted Pathloss Radio Map Prediction Data Competition. We propose a physics-based feature engineering approach combined with a U-Net architecture featuring ResNet-34 encoder and Atrous Spatial Pyramid Pooling (ASPP) module to reconstruct indoor pathloss maps from extremely sparse ground-truth samples. Our method transforms the three-channel input into eight physics-based channels incorporating free-space pathloss, cumulative transmittance losses, log-distance from the antenna, and a binary padding mask. Through a combination of geometric augmentations and multi-scale feature extraction, we achieve competitive performance across both uniform and strategic sampling scenarios. The model attains a weighted RMSE of 5.17 dB, while maintaining inference times of 65.2 ms per map including preprocessing—orders of magnitude faster than ray-tracing baselines.

*Index Terms*— Pathloss radio map prediction, deep learning, sparse measurements, U-Net.

## 1. INTRODUCTION

Recent advancements in machine learning have significantly accelerated the development of solutions to radio frequency (RF)-related challenges. These challenges encompass a broad range of tasks, including wireless device localization [1, 2, 3], environmental reconstruction [4, 5], and radio map prediction [6, 7, 8, 9]. A diverse range of neural architectures has been explored for these tasks, from convolutional neural networks (CNNs), which effectively model local spatial dependencies, to vision transformers, which offer enhanced capacity for modeling long-range interactions. These models have demonstrated promising performance across both indoor and outdoor scenarios, marking a significant step forward in the application of deep learning to RF signal processing. In this paper, we focus on the task of indoor pathloss radio map prediction, assuming access to known pathloss measurements at a limited number of spatial locations.

Accurate prediction of large-scale pathloss (PL) within buildings is essential for access point placement, channel-aware scheduling, and localization services [10, 11, 12]. Traditional approaches rely on either empirical models, which lack accuracy in complex indoor environments, or deterministic methods like ray tracing, which provide high fidelity but require hours of computation per map. The emergence of data-driven methods offers a promising middle ground, potentially achieving near ray-tracing accuracy with dramatically reduced computational cost.

Traditional analytical models often fall short in capturing complex propagation effects, requiring considerable manual tuning and empirical calibration. Recent studies, such as [13], highlight the superior performance of deep learning models for pathloss estimation, while [14] demonstrates that classical log-distance models are outperformed by learning-based methods, including neural networks and statistical regressors. Although statistical models provide rough estimates by assuming monotonic signal decay with distance, their predictions can be significantly improved using neural networks [15]. Furthermore, while ray-tracing methods deliver high accuracy, they are computationally expensive, making deep learning-based solutions a more efficient alternative [16].

The Sampling-Assisted Pathloss Radio Map Prediction Data Competition of IEEE MLSP 2025 [17] addresses a critical real-world scenario: reconstructing complete indoor radio maps from minimal field measurements. This reflects practical deployment constraints where exhaustive measurements are infeasible. Participants must predict pathloss values across entire building floors using only 0.02% to 0.5% of ground-truth samples, combined with environmental information encoded as RGB images representing material properties and distances.

Building on our prior work [8], where vision transformers (ViTs) were employed for the first indoor pathloss radio map prediction challenge [18], we adopt a different strategy in this study by leveraging CNNs. CNNs offer greater computational efficiency and are particularly well-suited for scenarios with limited training data. While large transformer-based models can effectively learn complex representations given sufficient data, our current setting requires careful and deliberate feature engineering. Under these constraints, CNNs demonstrate strong performance by utilizing the engineered features to produce accurate and reliable pathloss predictions.

Our solution employs deep learning with physics-based feature engineering to address the challenge of pathloss radio map prediction. We transform the provided input channels into an eight-channel representation that captures electromagnetic propagation physics, including free-space pathloss, cumulative transmittance losses, log-distance from the antenna, and a binary mask of padded pixels. This physics-based representation is processed through a U-Net architec-

ture with ResNet-34 encoder combined with an ASPP module. Our approach achieved the weighted RMSE of 5.17dB on the competition evaluation set, with inference time of $65.2 \pm 7.7$ ms per map including preprocessing on a single A6000 GPU.

The key contributions of this work include: (1) a physics-based feature engineering approach that significantly improves prediction accuracy, (2) an effective architecture combining U-Net with ASPP for indoor propagation modeling, (3) a comprehensive augmentation strategy that preserves physical relationships while improving generalization, and (4) a distance-weighted sampling approach for optimal measurement placement.

This paper is organized as follows: Section 2 reviews related works in pathloss prediction; Section 3 describes the challenge dataset and evaluation protocol; Section 4 details our methodology including feature engineering, network architecture, and training strategy; Section 5 presents our data augmentation strategy; Section 6 presents experimental results with ablation studies and computational analysis and then discusses key findings and limitations; and Section 7 concludes and summarizes our findings.

## 2. RELATED WORKS

A variety of approaches have been developed to address the challenge of pathloss prediction in both indoor and outdoor settings [10, 19]. Among these, convolutional encoder-decoder architectures have shown considerable promise, effectively modeling spatial dependencies necessary for accurate radio signal attenuation estimation [7, 20].

The method presented in [9] introduces a UNet-based architecture tailored for efficient radio map generation in urban environments, particularly accounting for the mobility of base stations and user equipment. Similarly, the work in [7] employs a SegNet-based model to address the complexities of outdoor pathloss estimation. In [21], the authors incorporate a line-of-sight (LoS) map as an additional input feature, representing a feature engineering strategy that proves especially valuable in scenarios where labeled data is scarce.

In the context of indoor radio map prediction, the top-performing solution of the first indoor pathloss prediction challenge [18] SIP2Net utilized a UNet-style architecture combined with a custom loss function to achieve superior results [22]. Another competitive method, IPP-Net [15], enhanced model performance through the integration of building-specific features such as the number of walls separating the transmitter from each spatial location. A notable alternative, TransPathNet [6], approaches indoor pathloss estimation using a two-stage framework, where an initial coarse prediction is subsequently refined by a secondary network to improve spatial resolution and signal-level accuracy.

The task of pathloss radio map prediction becomes particularly compelling when only sparse measurement data, typically obtained through field surveys, is available. In [23], the authors address this challenge using a meta-learning framework. Their approach involves partitioning the target area into a $100 \times 100$ grid and subsequently refining the pathloss predictions using sparse measurements. Similarly, the authors of [24] leverage synthetic datasets generated via ray-tracing simulations to pre-train a pathloss prediction model, which is then fine-tuned with limited real measurement data to enhance generalization.

Motivated by these studies, we introduce a U-Net-based convolutional encoder-decoder architecture tailored for indoor pathloss prediction with sparse supervision. To compensate for the limited availability of data, we incorporate manually engineered, physics-based features into the model's input representation, thereby guiding the learning process with domain-relevant priors.

## 3. CHALLENGE SETUP

### 3.1. Dataset Description

The organizers provide the Indoor Radio Map Dataset [25] generated using the Ranplan Wireless ray-tracer, a commercial-grade propagation simulator. The dataset exhibits significant diversity in building layouts, ranging from simple few-rectangle structures to complex multi-room environments with varying wall materials and thicknesses. The training partition contains 25 buildings, each with 50 transmitter locations simulated at three carrier frequencies (868 MHz, 1.8 GHz, and 3.5 GHz), yielding 3 750 radio maps. We train on the full multi-frequency set to leverage cross-frequency patterns. The test set comprises five previously unseen buildings (250 maps) evaluated only at 868 MHz. All maps use a 0.25 m grid, transmitter height 1.5 m, and path-loss range 13–160 dB. Each radio map is paired with a three-channel input image where **Channel 1** contains normal incidence reflectance coefficients (0 for air), **Channel 2** contains normal incidence transmittance coefficients (0 for air), and **Channel 3** contains the Euclidean distance from transmitter to each grid point in meters.

### 3.2. Competition Tasks

The workshop evaluates two sampling regimes. Task 1 (uniform sampling) draws measurement pixels uniformly at random, whereas Task 2 (strategic sampling) lets participants choose the measurement locations. In both cases the sample budget is $\lceil rWH/100 \rceil$, where $r \in \{0.02, 0.5\}$ is the sampling percentage and $W, H$ are the map dimensions.

### 3.3. Evaluation Protocol

Performance is measured using root mean-square error (RMSE) computed exclusively over unsampled locations:

$$\text{RMSE} = \sqrt{\frac{1}{\sum_{n \in \mathcal{T}}(W_n H_n - |\mathcal{S}_n|)} \sum_{n \in \mathcal{T}} \sum_{i,j} \mathbb{1}_{\{(i,j) \notin \mathcal{S}_n\}} (\tilde{P}_L^{(n)}(i,j) - P_L^{(n)}(i,j))^2}$$

where $\mathcal{T}$ denotes the test set, $n$ indexes individual radio maps, $(W_n, H_n)$ are the map dimensions, $\mathcal{S}_n$ contains the sampled locations, $\mathbb{1}_{\{.\}}$ is the indicator function, $\tilde{P}_L^{(n)}(i,j)$ is the predicted pathloss at pixel $(i,j)$, and $P_L^{(n)}(i,j)$ is the ground-truth value from ray-tracing.

The final leaderboard score combines four sub-task scores:

$$\text{FinalScore} = 0.3\,(\text{RMSE}_{\text{T1A}} + \text{RMSE}_{\text{T1B}}) + 0.2\,(\text{RMSE}_{\text{T2A}} + \text{RMSE}_{\text{T2B}})$$

This weighting scheme emphasizes Task 1 (60%) over Task 2 (40%), reflecting the importance of strong baseline performance before optimization through strategic sampling.

## 4. METHODOLOGY

### 4.1. Physics-Based Feature Engineering

We explicitly incorporate electromagnetic wave propagation principles, including free-space path loss, material interactions, and cumulative attenuation effects. Our approach transforms the basic three-channel input into a rich eight-channel representation crucial for achieving competitive performance. As part of size normalization, each input is uniformly scaled and zero-padded to 640×640, yielding a binary mask of padded pixels as one feature channel.

### 4.1.1. Free-Space Pathloss

The free-space pathloss (FSPL) provides a fundamental baseline for signal attenuation. As described in [26], the FSPL is given by:

$$\text{PL}_{\text{FS}} = 20\log_{10}(d) + 20\log_{10}(f) - 27.55$$

where $d$ is the transmitter-receiver distance in meters (clamped to a minimum of 0.125 m to avoid singularities), $f$ is the carrier frequency in MHz, and the constant 27.55 accounts for the specific choice of units.

### 4.1.2. Cumulative Transmittance Loss

Beyond free-space propagation, indoor environments introduce additional losses through wall penetration. We developed a Numba-accelerated algorithm that traces rays from the transmitter to each grid point, accumulating transmittance losses:

$$\text{Loss}_{\text{trans}}(x,y) = \min_{\theta} \sum_{i \in \text{walls}(\theta)} T_i$$

where $T_i$ represents the transmittance loss of wall $i$ along ray direction $\theta$. The algorithm considers multiple ray angles (360×128) and selects the path with minimum cumulative loss, approximating the dominant propagation path.

Figure 1 shows the resulting transmittance loss map, clearly delineating regions separated by walls and capturing the additional attenuation from building materials.

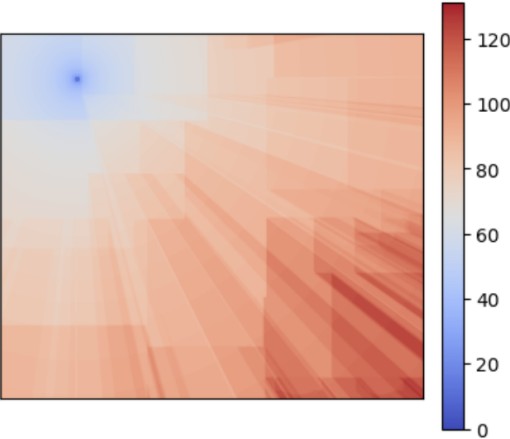

**Fig. 1**. Cumulative transmittance loss feature showing additional attenuation through walls. The sharp boundaries correspond to wall locations, with higher losses in regions requiring penetration through multiple walls.

### 4.1.3. Complete Feature Stack

Our final eight-channel input comprises:
- Original reflectance coefficients
- Original transmittance coefficients
- Free-space pathloss
- Frequency channel (log-normalized)
- Cumulative transmittance loss
- The sparse measurements
- Log-distance from the antenna
- Binary mask of padded pixels

This physics-based representation enables the network to leverage domain knowledge about electromagnetic propagation while learning complex, environment-specific variations, and also facilitates physics-based augmentations that maintain propagation consistency.

### 4.2. Network Architecture

We employ a U-Net architecture enhanced with modern deep learning components, as illustrated in Figure 2. The encoder uses a randomly initialized ResNet-34 backbone, modified to accept **8** input channels. The deep residual architecture provides several advantages: strong feature extraction capabilities from pretrained weights, effective gradient flow through residual connections, and multi-scale feature hierarchy from different stages.

At the bottleneck, we integrate an Atrous Spatial Pyramid Pooling (ASPP) module [27] with parallel branches comprising a 1×1 convolution for local features, 3×3 convolutions with dilation rates 6, 12, 18, and global average pooling for scene-level context. This multi-scale processing proves crucial for capturing both local wall interactions and long-range propagation effects. The different dilation rates allow the network to aggregate information across various spatial extents without losing resolution.

The decoder follows standard U-Net design with skip connections from corresponding encoder layers. The final layer produces single-channel pathloss predictions in dB.

## 5. DATA AUGMENTATION STRATEGY

We apply geometric and physics-based augmentations that preserve physical relationships while increasing training data diversity:

**Rotation augmentation** includes both continuous rotations in $[-30, 30]$ and discrete cardinal rotations. For continuous rotations, we update:
- Antenna position using bilinear interpolation
- All feature channels using appropriate interpolation

**Distance scaling** by factors in $[1/1.5, 1.5]$ requires adjusting the FSPL:

$$\Delta\text{PL} = 20\log_{10}(s)$$

where $s$ is the scale factor. This maintains physical consistency between distance and pathloss.

**Synthetic wall insertion** augments each training map by adding a few extra axis-aligned walls. We draw one to $k$ random columns and rows (with $k$ uniformly sampled between one and an upper bound proportional to the current wall density) and assign each new wall a transmittance coefficient chosen uniformly from the integer range [2,18]. The reflectance channel is left unchanged; we add the selected coefficients to the transmittance channel, recompute the cumulative-transmittance feature, and update the path-loss label by adding the resulting extra loss in dB. This way the network learns to handle maps with denser interior layouts.

**Flipping operations** (horizontal/vertical) are applied to increase spatial diversity.

All geometric augmentations are applied with 50% probability during training, while wall insertion is applied independently, significantly improving generalization to unseen building geometries and structural variations.

### 5.1. Training Configuration

Our training strategy employs several techniques to ensure robust convergence: **Data splitting:** out of 25 buildings 22 are randomly

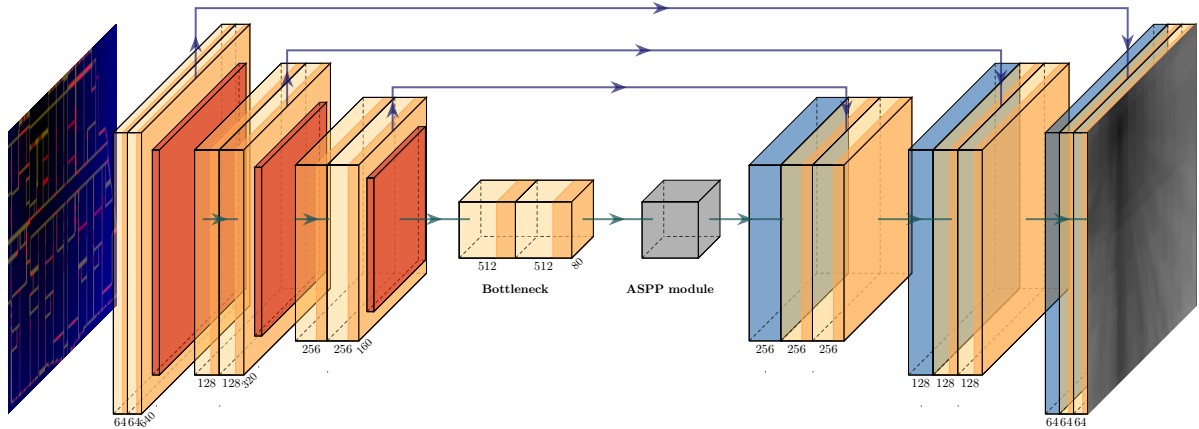

**Fig. 2**. Model architecture.

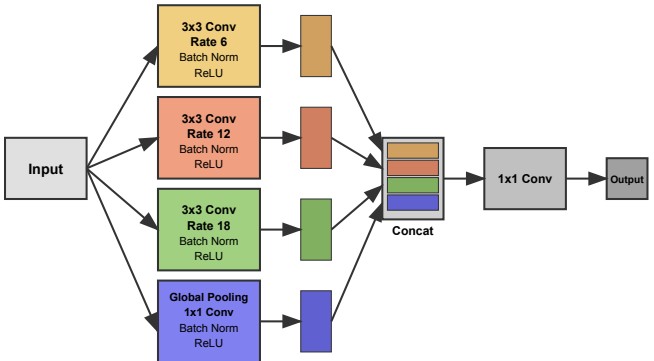

**Fig. 3**. The architecture of the ASPP module of the network.

chosen for training, and 3 for validation, maintaining complete building separation to assess generalization. **Multi-frequency training:** We trained on the complete dataset spanning all three frequencies (868 MHz, 1.8 GHz, 3.5 GHz), despite testing only on 868 MHz. This multi-frequency training provides several benefits: increased training data volume (3,750 vs. 1,250 samples), implicit regularization through frequency diversity, and learning of frequency-invariant propagation patterns. The frequency information is explicitly provided as an input channel, enabling the model to learn frequency-dependent adjustments while leveraging shared propagation physics across frequencies. **Optimization details:** We use the Adam optimizer with $\beta_1 = 0.9$, $\beta_2 = 0.999$ and a constant learning rate of $3 \times 10^{-4}$. **Loss function:** MSE computed only over unsampled locations:

$$\mathcal{L} = \frac{1}{|\mathcal{U}|} \sum_{(i,j) \in \mathcal{U}} (\hat{y}_{ij} - y_{ij})^2$$

where $\mathcal{U}$ represents unsampled grid points. **Implementation details:** We use batch size 32 on dual NVIDIA DGX A100 GPUs and train for 2000 epochs with early stopping. The input resolution is 640×640 (resized with nearest neighbor for materials, bilinear for distances). For **variable sparsity sampling**, during training, for each map we generate the sparse-measurement channel by either providing no measurements (0%) with probability 0.25, or with probability 0.75 picking $r \sim \mathcal{U}(0, 0.005)$ and sampling that fraction of locations. Training a single model per task over this full 0.02–0.5% range yields a slightly better validation error with half

the number of models versus fitting separate models at fixed 0.02% and 0.5%.

## 5.2. Strategic Sampling Algorithm

For Task 2, we developed a distance-weighted sampling strategy based on the observation that pathloss uncertainty typically increases with distance from the transmitter.

The algorithm first computes distance-weighted sampling probabilities for each pixel location $(i, j)$ as:

$$P(i,j) = \frac{d_{i,j}^{\alpha}}{\sum_{k,l} d_{k,l}^{\alpha}}$$

where $d_{i,j} = \sqrt{(i - y_{ant})^2 + (j - x_{ant})^2}$ is the Euclidean distance from the transmitter position and $\alpha = 2$. This weighting favors locations farther from the transmitter, where pathloss predictions are typically less reliable.

To balance information gain with spatial diversity, we employ a two-stage selection process. First, we oversample candidate locations by drawing $10N$ points (where $N$ is the target number of samples) according to the distance-weighted probabilities. From these candidates, we apply greedy selection with spatial separation constraints, requiring selected points to maintain minimum separation of $0.5\sqrt{HW/N}$ pixels, where $H$ and $W$ are the map dimensions. This separation distance is designed to ensure reasonable spatial coverage across the building.

If the spatial constraint prevents selection of sufficient points, remaining slots are filled by randomly sampling from all unselected locations using the original distance-weighted probabilities. This ensures we always obtain the required number of measurements.

This approach aims to balance exploration of high-uncertainty regions (far from transmitter) with adequate spatial coverage throughout the building. Figure 4 shows an example of the resulting sampling pattern, illustrating how points concentrate in regions distant from the transmitter while maintaining spatial separation. We observe modest to no gains from strategic sampling using this algorithm, suggesting that the uniform sampling baseline is already quite effective for the given sampling densities.

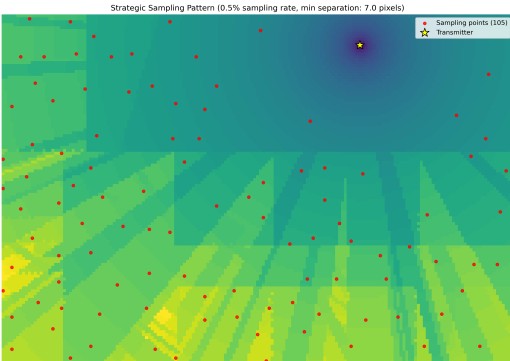

**Fig. 4**. Example of strategic sampling pattern showing selected measurement locations (red dots) overlaid on a building layout. The sampling concentrates in regions distant from the transmitter (yellow star) while maintaining spatial separation constraints.

---

**Algorithm 1:** Distance-weighted greedy sampler for $N$ measurements

---

**Input** : Radio-map grid size $H \times W$;
        Transmitter coordinates $(y_{\text{ant}}, x_{\text{ant}})$;
        Desired sample count $N$.
**Output:** Set $\mathcal{S}$ of $N$ pixel indices

**Step 1: probability map.** For every pixel $(i, j)$ compute

$$d_{i,j} \leftarrow \sqrt{(i - y_{\text{ant}})^2 + (j - x_{\text{ant}})^2}, \quad P(i,j) \leftarrow \frac{d_{i,j}^2}{\sum_{k,\ell} d_{k,\ell}^2}$$

**Step 2: candidate pool.** Draw $10N$ indices from $P$ to form $\mathcal{C}$; sort $\mathcal{C}$ by decreasing $P$;

$\mathcal{S} \leftarrow \emptyset$;
$d_{\min} \leftarrow 0.5\sqrt{HW/N}$ ; // spacing threshold
**foreach** $(i, j) \in \mathcal{C}$ **do**
   **if** $|\mathcal{S}| = N$ **then**
      └ **break**
   **if** $\forall (u, v) \in \mathcal{S} : \|(i, j) - (u, v)\|_2 \geq d_{\min}$ **then**
      └ $\mathcal{S} \leftarrow \mathcal{S} \cup \{(i, j)\}$ ; // accept

**while** $|\mathcal{S}| < N$ **do**
   Draw $(i, j)$ from $P$ (no spacing check);
   $\mathcal{S} \leftarrow \mathcal{S} \cup \{(i, j)\}$
**return** $\mathcal{S}$ ;                 // $\mathcal{O}(N^2)$ worst-case

---

## 6. RESULTS AND ANALYSIS

Table 1 presents our performance across all competition tasks. The model demonstrates strong results at both sampling rates, with expected improvement as sampling density increases.

Several key observations emerge:

- **Sampling rate impact:** Increasing sampling from 0.02% to 0.5% reduces RMSE by 35–45%, demonstrating the value of even very few additional measurements.
- **Strategic vs uniform:** Our strategic sampling shows modest improvements at 0.5% (9.5% reduction) but performs slightly worse at 0.02%, suggesting the distance-based strategy is most effective with sufficient samples.
- **Consistency:** Similar performance patterns across tasks indicate robust generalization.

**Table 1**. Competition results showing RMSE (dB) for each task and sampling rate

| Task | RMSE (dB) 0.02% | RMSE (dB) 0.5% | Weight | Contribution |
|---|---|---|---|---|
| Task 1 (Uniform) | 6.42 | 4.01 | 0.3 | 3.15 |
| Task 2 (Strategic) | 6.65 | 3.63 | 0.2 | 2.04 |
| | | | Final Score: | **5.17** |

### 6.1. Ablation Studies

To understand the contribution of different components, we conducted systematic ablations evaluating various architectural choices and feature combinations. Table 2 presents the complete ablation results on Task 1 with 0.5% sampling.

**Table 2**. Ablation results on Kaggle for Task 1 (0.5% sampling)

| ASPP | FSPL | Trans. Loss | RMSE (dB) | $\Delta$ vs Best |
|---|---|---|---|---|
| ✓ | ✓ | ✓ | **4.05** | – |
| ✗ | ✓ | ✓ | 4.32 | +0.27 |
| ✗ | ✗ | ✓ | 4.35 | +0.30 |
| ✓ | ✗ | ✓ | 4.36 | +0.31 |
| ✓ | ✓ | ✗ | 4.81 | +0.76 |
| ✗ | ✗ | ✗ | 4.86 | +0.81 |
| ✓ | ✗ | ✗ | 4.88 | +0.83 |
| ✗ | ✓ | ✗ | 5.07 | +1.02 |

**Key take-aways (brief)**

- Best model uses *all* cues (ASPP + FSPL + Trans.) and achieves 16.4dB.
- Transmittance loss is pivotal: removing it costs $\approx$2–2.6dB even when other cues remain.
- FSPL and ASPP are complementary—each alone offers minor gains, but together with transmittance loss they cut error by an extra 2.3dB.

### 6.2. Discussion

Our ablations (Table 2) confirm that cumulative transmittance loss is the single most critical cue—dropping it raises the error up to 2.6dB—while FSPL and ASPP each yield roughly 0.5–2.3dB improvements, highlighting their complementary roles. Training one model per task over a variable sparsity range (0.02–0.5%) improved Kaggle validation by $\approx$0.1–0.2dB versus fixed-0.02% and 0.5% models and halved the number of models. Strategic sampling cuts RMSE by $\sim$0.2dB at 0.5% but slightly degrades performance at 0.02%, likely due to density-dependent benefits.

## 7. CONCLUSION

We presented a novel approach to indoor pathloss prediction from sparse samples, combining physics-based feature engineering with modern deep learning architectures. Our solution achieves competitive accuracy (5.17 dB RMSE) while maintaining real-time inference capabilities suitable for practical deployment. Key contributions include physics-based feature engineering that explicitly encodes propagation mechanisms, effective integration of U-Net with ASPP for multi-scale indoor propagation modeling, comprehensive augmentation strategy preserving physical relationships, and distance-weighted strategic sampling balancing exploration and coverage. The strong performance with minimal sampling (0.02-0.5%)

demonstrates the feasibility of accurate radio map prediction with limited measurements, potentially enabling cost-effective deployment of AI-assisted network planning tools. Our code and trained models will be released upon publication at our GitHub repository to facilitate reproducible research and practical adoption of these techniques.

## 8. ACKNOWLEDGEMENTS

This work was supported by funding under the bilateral agreement between CNR (Italy) and HESC MESCS RA (Armenia) as part of the DeepRF project for the 2025–2026 biennium, and by the HESC MESCS RA grant No. 22rl-052 (DISTAL).

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
