# OpenReview forum: "U-Net for Indoor Pathloss Prediction from Sparse Measurements with Physics-Informed Features"
_IEEE.org/MLSP/2025_SA_Radio_Map_Prediction_Challenge — SA Radio Map Prediction Challenge at MLSP 2025 Oral_

### Official Review · Reviewer_s3Cs · 2025-06-05
**The manuscript proposes a physically-aware feature engineering approach combined with a U-Net architecture with ResNet-34 encoder and spatial pyramid pooling (ASPP) module for effective radio map estimation.**

**Rating:** 7
**Confidence:** 4

**Review:**

Overall, it's a good job. However, on some minor technical issues, I make the following comments:
1.Please check the accuracy of the wRMSE given in the manuscript, which is inconsistent with the conclusions shown on the contest webpage.
2.References are too redundant, consider removing some that are not very relevant and old.
3.An error symbol appears in the first paragraph of chapter 2.
4.The manuscript considers a wide range of input features, e.g., sections 4.1.1 and 4.1.2. However, the validity does not and proves that it would be interesting to add this section for ablation experiments, if it is executable of course.
5.Essentially, radio map reconstruction using the ASPP module, which is derived from the Deeplabv3 model and should be referenced rather than subsequent improved methods, utilizes the larger sense field of dilated convolution.
6.The sampling strategy in section 5.1 is difficult to understand, if possible describe or show the sampling results in detail.

---

### Official Review · Reviewer_AoEM · 2025-06-06
**Review for "U-Net for Indoor Pathloss Prediction from Sparse Measurements with Physics-Informed Features"**

**Rating:** 7
**Confidence:** 4

**Review:**

This paper proposes a physics-aware feature engineering method combined with a U-Net architecture featuring a ResNet-34 encoder and an ASPP module to reconstruct indoor pathloss maps from extremely sparse ground-truth samples. The proposed method designs up to eight physics-informed channels, balancing accuracy and efficiency, making it well-suited for practical wireless network planning applications. Below are several revision suggestions that should be carefully addressed:
1. The citation numbering in the Introduction is somewhat confusing. For example, in the sentence “Accurate prediction of large-scale pathloss (PL) within buildings is essential for access point placement, channel-aware scheduling, and localization services [15, 16, 17, 2, 3, 4, 5],” splitting the citations (e.g., writing “localization services [2, 3, 4, 5]”) would improve readability. Please review the entire paper to ensure references are cited in appropriate and consistent order.

2. In Section 2 "Related Works", some citations are incomplete, such as “[?, 24]”. Additionally, references [8], [14], [20], and [22] all have officially published conference versions. Please avoid citing unreviewed arXiv versions when a peer-reviewed version is available.

3. The manuscript includes many sections formatted using \item, which negatively impacts the flow of the text. It is recommended that the authors revise the formatting and reorganize the content for smoother narrative flow.

4. In Section 4.1 “Physics-Aware Feature Engineering”, the sentence “Our approach transforms the basic three-channel input into a rich eight-channel representation crucial for achieving competitive performance.” appears twice consecutively. Please check the manuscript carefully to avoid such typographical repetitions.

5. Please update the RMSE results in the manuscript to ensure consistency with those published on the official competition website.

6. Table 2 presents evaluation results using the MSE metric from Kaggle. Please convert these to RMSE values to maintain consistency across all performance metrics in the paper.

7. The terms “physics-aware” and “physics-informed” are used interchangeably throughout the paper. Please ensure consistent usage of terminology for clarity and coherence.

---

### Official Review · Reviewer_Vbyt · 2025-06-09
**Review for Submission 8**

**Rating:** 6
**Confidence:** 3

**Review:**

The paper introduces an effective feature engineering pipeline based on electromagnetic propagation principles, which achieves competitive RMSE scores (5.19 dB weighted), outperforming many baselines in both uniform and strategic sampling tasks. The inference time of ~100ms per map, including preprocessing, is practical for real-time applications. Below are some suggestions:

1. Consecutive citations like [1], [2], [3] should be merged as [1–3]. Non-consecutive references should be in a single bracket and sorted numerically, e.g., [1, 3, 5] instead of [5][1][3]. Also, there are missing citations [?] in section 2.
2. In the abstract, he paper claims a final weighted RMSE of 5.19 dB in the abstract and results section. However, according to the official leaderboard of the MLSP 2025 Sampling-Assisted Pathloss Radio Map Prediction Challenge, the final score is 5.17 dB. Similar problem occurs in Table 1. The authors should ensure consistency between the manuscript and official evaluation sources.
3. The paper reports an inference time of approximately 100 ms per map, but this appears to be an estimate rather than a rigorously measured value. No information is provided about the hardware used (e.g., GPU/CPU model), batch size, or whether preprocessing is included in the inference time.
4. The paper presents detailed quantitative results (e.g., RMSE scores) at the end of the Introduction. While summarizing key outcomes is useful, such detailed reporting is better suited for the Results section. The Introduction should focus on setting up the problem, motivating the approach, and summarizing contributions qualitatively.
5. The manuscript overuses itemized lists (\item) in multiple sections. While such formatting improves clarity in some cases, excessive use makes the text feel fragmented and impairs narrative flow. The authors are encouraged to restructure certain lists into coherent paragraphs for a more polished and scholarly presentation.
6. The formulas in the paper use informal and ambiguous variable names such as Final. These names are not self-explanatory and do not follow standard mathematical or physical notation. Authors should use formal variable names and provide proper definitions for all terms involved in the equation.
7. Strategic sampling brings minor or no improvement according to results provided by the author. It's highly recommended that the authors explain the reason behind it.
8. The paper uses the classic free-space pathloss equation with a constant term (−27.55) but does not cite its origin. A proper reference should be included to acknowledge its source and clarify the underlying assumptions.
9. The final paragraph discusses general trends in indoor wireless networks and the potential value of AI-assisted tools. While relevant, this content is better suited for the Introduction as it frames the motivation rather than summarizes findings. The Conclusion should focus on key technical contributions, observed performance, and limitations or future directions.

---

### Official Review · Reviewer_eikM · 2025-06-09
**Review for "U-Net for Indoor Pathloss Prediction from Sparse Measurements with Physics-Informed Features"**

**Rating:** 8
**Confidence:** 4

**Review:**

1) Please revise Section 5.1.3. A simpler explanation without so much sophisticated terminology would be better. For example, it is unclear what the authors mean by "residual distribution" or "oversampled candidates."
2) It would be better to include the link to the repository in a section other than "Conclusions," such as "Abstract," "Introduction," "Methodology," or "Results."
3) The "Acknowledgements" and "References" headings do not seem to adhere to the MLSP paper specifications.
4) "100ms", spacing between numbers and units
5) It is probably not necessary for periods to be placed after the enumerated authors' affiliations at the beginning.